# Improving Biologics’ Effectiveness in Clinical Oncology: From the Combination of Two Monoclonal Antibodies to Oligoclonal Antibody Mixtures

**DOI:** 10.3390/cancers13184620

**Published:** 2021-09-15

**Authors:** Christel Larbouret, Laurent Gros, André Pèlegrin, Thierry Chardès

**Affiliations:** 1Institut de Recherche en Cancérologie de Montpellier (IRCM), Institut Régional du Cancer de Montpellier (ICM), Inserm U1194, Université de Montpellier, 34298 Montpellier, France; laurent.gros@inserm.fr (L.G.); andre.pelegrin@inserm.fr (A.P.); thierry.chardes@inserm.fr (T.C.); 2Centre National de la Recherche Scientifique (CNRS), 75016 Paris, France

**Keywords:** cancer, antibody, biologic, immunotherapy, combination, oligoclonal, mixture

## Abstract

**Simple Summary:**

The approval of the two antibody combinations trastuzumab/pertuzumab and ipilimumab/nivolumab in oncology has paved the way for novel antibody combinations or oligoclonal antibody mixtures to improve their efficacy in cancer. The underlying biological mechanisms and challenges of these strategies will be discussed using data from clinical trials listed in databases. These therapeutic combinations also lead to questions on how to optimize their formulation and delivery to induce a therapeutic polyclonal response in patients with cancer.

**Abstract:**

Monoclonal antibodies have revolutionized the treatment of many diseases, but their clinical efficacy remains limited in some other cases. Pre-clinical and clinical trials have shown that combinations of antibodies that bind to the same target (homo-combinations) or to different targets (hetero-combinations) to mimic the polyclonal humoral immune response improve their therapeutic effects in cancer. The approval of the trastuzumab/pertuzumab combination for breast cancer and then of the ipilimumab/nivolumab combination for melanoma opened the way to novel antibody combinations or oligoclonal antibody mixtures as more effective biologics for cancer management. We found more than 300 phase II/III clinical trials on antibody combinations, with/without chemotherapy, radiotherapy, small molecules or vaccines, in the ClinicalTrials.gov database. Such combinations enhance the biological responses and bypass the resistance mechanisms observed with antibody monotherapy. Usually, such antibody combinations are administered sequentially as separate formulations. Combined formulations have also been developed in which separately produced antibodies are mixed before administration or are produced simultaneously in a single cell line or a single batch of different cell lines as a polyclonal master cell bank. The regulation, toxicity and injection sequence of these oligoclonal antibody mixtures still need to be addressed in order to optimize their delivery and their therapeutic effects.

## 1. Introduction

In the 19th century, the pioneering work of Shibasaburo Kitasato and Emil von Behring in Germany and Emile Roux in France paved the way for serotherapy. This treatment is based on the use of sera that originate from previously immunized animals or humans and contain pathogen-specific antibodies as the active substance. César Milstein and Georges Köhler revolutionized this concept by inventing the lymphocyte hybridization technique that led to the development of a new pharmacological class of biologics called “monoclonal” antibodies (mAbs). However, partial and short-lived responses, often associated with resistance phenomena (extensively studied in basic research), limit the clinical efficacy of mAbs. To overcome these obstacles, mAb combinations, most often evaluated separately, and oligoclonal antibody cocktails, considered as a single biologic, have been developed. Indeed, the immune system has naturally evolved to generate a polyclonal humoral response to optimize its ability to fight diseases, rather than the monoclonal strategy proposed by the currently approved antibody biologics. In this review, we first describe pre-clinical studies showing the potential of co-targeting tumor and/or immune checkpoint molecules with antibodies in oncology. Antibody mixtures can be made of antibodies against the same target (i.e., homo-combinations) or against different targets (i.e., hetero-combinations). The approval of two therapeutic antibody combinations, trastuzumab/pertuzumab and ipilimumab/nivolumab, validated this concept of “mimicking” the polyclonal humoral immune response for cancer treatment. We then list the antibody combinations that are currently tested in phase II and III clinical trials. Finally, we discuss how the technical improvements for the reproducible manufacturing of oligoclonal antibody mixtures, in which each antibody is selected on the basis of specific criteria (e.g., epitope specificity, affinity or intrinsic biological activity), now allow the natural polyclonal humoral immune response to be mimicked, paving the way for 21st century serotherapy.

## 2. Homo-Combinations and Hetero-Combinations of Antibodies in Preclinical Studies

### 2.1. Tumor Co-Targeting in Oncology

Around the year 2000, the notion of homo-combination of antibodies, involving distinct epitopes on the same receptor, was pioneered by Yosef Yarden (Weizmann Institute, Israel) and then by other research groups. For instance, homo-combinations of antibodies against epidermal growth factor receptor (EGFR) [1,2,3,4,5], human epidermal growth factor receptor-2 (HER2) [6,7,8,9] or hepatocyte growth factor (HGF) receptor (i.e., cMET) [10,11] induce synergistic anti-tumor activity due to accelerated degradation of the targeted receptors and enhanced antibody-dependent cell-mediated cytotoxicity (ADCC) (Figure 1). Moreover, these antibody combinations bypass the resistance to treatment induced by monotherapy with cetuximab (anti-EGFR mAb) in colorectal cancer [12] and with an anti-cMET antibody in gastric cancer [13]. They also maintain anti-tumor activity despite the presence of EGFR extracellular domain mutations that might impair antibody binding [14].

In 2007, our team demonstrated that the hetero-combination of antibodies against EGFR and HER2, two functionally collaborating receptors (Figure 1), has a higher anti-tumor effect by promoting ADCC, by reducing the expression of these receptors and homodimer formation [16,17,18] and also by inhibiting intracellular signaling pathways [19]. This preclinical work, confirmed by other research groups [20,21,22], led to the initiation of the THERAPY phase I/II clinical trial in patients with metastatic pancreatic cancer who progressed on gemcitabine. This trial showed that the combination of cetuximab and trastuzumab (targeting EGFR and HER2, respectively) stabilizes the disease in 27% of patients, without objective response but with a positive correlation between skin toxicity and progression-free survival [23]. The clinical trial was discontinued because of high toxicity, highlighting the need to rethink the active dose when using antibody combinations. The idea behind the administration of this hetero-combination was to avoid compensatory signaling phenomena related to the targeting of a single receptor. It was then extended to the dual targeting of EGFR and HER3 in cetuximab- and osimertinib-resistant tumors [12,24]. Such antibody hetero-combinations can also include antibodies against a ligand, for instance vascular endothelial growth factor (VEGF) or HGF, and a receptor (Figure 1), to target both the tumor microenvironment and a tumor-specific receptor [25,26,27].

Finally, oligoclonal cocktails of three [28,29] or six [30,31] antibodies against EGFR, HER2 and HER3 have an increased anti-tumor effect in experimental models, with blockade of the underlying extracellular signal-regulated kinase (ERK) and protein kinase B (AKT) signaling pathways and accelerated receptor degradation. The six-mAb cocktail PanHER (Sym013) demonstrated strong efficiency in gemcitabine-sensitive and also in chemotherapy-resistant pancreatic cancer by downregulating these three receptors [32]. Homo- or hetero-combinations of antibodies against CD20, CD22 or CD52 expressed by B lymphocytes (and T cells for CD52) have also been proposed for blood malignancies [33].

### 2.2. Co-Targeting of Immune Checkpoint Molecules (ICM): Awakening the Immune System

The importance of immune checkpoints, such as cytotoxic T lymphocyte-associated protein 4 (CTLA-4) and programmed cell death 1 (PD-1), in modulating the anti-tumor T-cell response has been highlighted by the awarding of the 2018 Nobel Prize in Physiology or Medicine to James Allison and Tasuku Honjo. The understanding of their roles in regulating lymphocyte activation and tumor immune escape led to the development of antibody combinations against molecules of this functional family that are classified in co-inhibitory molecules (that must be blocked) and co-activating molecules (that must be stimulated). In 2010, it was shown that the hetero-combination of anti-CTLA-4/-PD-1 blocking antibodies (Figure 1) displays an increased anti-tumor efficacy in mouse models of colorectal cancer and melanoma. This effect is characterized by increased infiltration of cytotoxic T cells and inhibition of regulatory T cells and suppressive myeloid cells [34]. Other hetero-combinations of antibodies targeting the ICMs PD-1 and 4-1BBL (also known as tumor necrosis factor receptor superfamily member 9 or CD137), PD-1 and lymphocyte-activating gene 3 (LAG3) [35], or CD137 and T-cell immunoglobulin and mucin-domain containing-3 (TIM-3) [36], among others, have been proposed to modulate the immune response in cancers. The link between the expression or overexpression of PD-L1 (the ligand of PD-1) and the efficacy of anti-HER2 antibodies in some patients [37,38,39] led to the testing of the combination of tumor-targeting mAbs (TTmAbs), for instance against HER family members, and of antibodies against ICMs in some cancers.

### 2.3. Pre-Clinical Studies to Understand the Mechanisms of Tumor-Targeting Antibodies in Combination with Immune Checkpoint Blockade

The first clinical successes with anti-CD20, anti-HER2 or anti-EGFR TTmAbs were mainly attributed to the interruption of their respective signaling pathways or their ability to induce ADCC; however, several data also suggested an essential role for the innate and adaptive immunity in the therapeutic outcome. Unfortunately, the use of these naked antibodies as monotherapy in advanced solid tumors, such as breast cancer, metastatic colorectal cancer and head and neck squamous cell carcinoma, results in a high proportion of tumors displaying primary and acquired resistance, and relatively low lasting therapeutic response rates. This suggested that TTmAbs should be associated with anti-ICM antibodies to obtain synergistic effects and sustained antitumor activity. Many clinical trials have been set up in recent years, but with variable success, depending on the cancer type and drugs used. To optimize these combinatorial approaches, preclinical animal models must be developed to better characterize and understand the mechanisms implicated in their effects.

We and others recently demonstrated, using several immunocompetent mice models of solid and hematological tumors, that TTmAbs can overcome immune tolerance and induce the development of an adaptive immune memory, leading to long-lasting effects [40,41,42,43]. It is now clear that TTmAbs have immunomodulatory effects via the Fc fragment, through the recruitment of antigen-presenting cells at the tumor site, better antigen presentation and stronger adaptive immunity with consequences for both the memory cytotoxic and humoral responses [44,45]. Therefore, antitumor therapeutic approaches in which TTmAbs and anti-ICM antibodies are combined to reinforce this antitumor response are interesting for awakening the exhausted antitumor immune response and to reach long-term remission. We demonstrated in the mouse B16F10 melanoma model that anti-PD1 antibodies synergize with the TA99 mAb against TYRP-1 expressed at the surface of malignant melanocytes. In mice treated with this combination, CD8^+^ T cells, natural killer (NK) and γδ T cells with cytolytic activity were increased as well as plasma antitumor IgGs, leading to better overall survival [40]. Similar results were described recently by another group using the TA99 mAb with the anti-CTLA-4 mAb or the agonist anti-CD137 mAb in the B16F10 mouse melanoma model [46]. In both cases, treatment with the TA99 mAb resulted in an increased expression of the secondary targets CTLA-4, PD1 or CD137, leading to an optimal combinatorial effect of the anti-PD1 or anti-CD137 mAbs.

In the past ten years, other groups have shown that TTmAbs in combination with anti-ICM antibodies might have synergistic effects on the host adaptive anti-immune response and on tumor eradication. For example, anti-angiogenic therapy using anti-VEGFR antibodies can elicit or enhance the anti-tumor immune response, and reciprocally, the immune system can support angiogenesis. Yasuda et al. demonstrated, in a mouse model of colorectal cancer, that the simultaneous blockade of PD1 and VEGFR2 induces a synergistic anti-tumor effect. This might occur through different mechanisms that may not be mutually exclusive [47]. More recently, Allen et al. investigated the efficacy of the anti-PDL-1 and anti-VEGFR2 antibody combination in mice bearing pancreatic neuroendocrine tumor, mammary carcinoma or glioblastoma [48]. They found that the anti-VEGFR2 antibody treatment is associated with lymphocyte homing into the tumor, whereas the anti-PD-L1 antibody induces activation of infiltrated CD4^+^ and CD8^+^ T cells that produce IFNγ. Another study showed that in head and neck cancer, the anti-EGFR mAb cetuximab combined with an anti-CD137 agonist antibody leads to tumor regression and prolonged survival. This might be dependent on enhanced NK cell degranulation and cytotoxicity, dendritic cell presentation and tumor antigen cross-presentation [49]. Moreover, it is known that blocking VEGF induces ICM expression, and that the combination of anti-PD-L1 and anti-VEGF antibodies synergistically suppresses tumor growth in small cell lung cancer [50]. Another example illustrates the capacity of TTmAbs to modify the tumor microenvironment and increase the efficacy of anti-ICM antibodies. Indeed, the anti-HER2 mAb trastuzumab can increase IFNγ production through the MyD88 pathway, and IFNγ induces PD-L1 expression on tumor cells. Consequently, anti-PD1/anti-PD-L1 antibodies can strengthen the antitumor activity of anti-HER2 TTmAbs [51].

The CD47 ligand and its receptor SIRPα are other ICMs targeted by antibodies in several clinical trials in combination with TTmAbs. In mouse models of hematological or solid cancer, rituximab synergizes with the humanized anti-CD47 antibody HU5F9-G4 to promote phagocytosis and to eliminate non-Hodgkin lymphoma and solid tumors in xenografted mice [52]. Interestingly, the combination of HU5F9-G4 with cetuximab or panitumumab (anti-EGFR mAbs) reduces tumor burden more than any of the monotherapies in immunodeficient mice harboring patient-derived xenografts [53].

Altogether, these data show that TTmAbs can modulate the tumor microenvironment (vasculature, cytokine profiles, innate immunity activation, increase of the adaptive immune repertoire) and therefore reinforce the antitumor response of anti-ICM antibodies to achieve tumor regression. However, drug combination protocols are complex and require data obtained in preclinical studies to find the optimal conditions for the combined delivery (administered doses, concomitant vs. sequential administration, formulations, pharmacokinetics). Understanding the mechanisms involved in their synergistic effects will allow optimal clinical therapeutic approaches to be developed to counteract treatment resistance in patients with cancer [54].

## 3. The Initial Clinical Proof of Concept about Antibody Combinations

In 2012, the first TTmAb homo-combination (the anti-HER2 mAbs trastuzumab and pertuzumab) was approved with docetaxel for the treatment of patients with HER2-amplified metastatic breast cancer. The phase III trial CLEOPATRA (*n* = 808 patients) found a mean progression-free survival of 18.7 months in the antibody homo-combination + docetaxel arm compared with 12.4 months in the trastuzumab alone + docetaxel arm [55]. Cardiac toxicity was comparable in the two arms. This antibody homo-combination with docetaxel was subsequently approved for neo-adjuvant treatment of newly diagnosed patients (APHINITY and NeoSphere trials [56,57]). However, the NeoSphere trial reported increased toxicity in the trastuzumab/pertuzumab + docetaxel arm. Recently, a subcutaneous formulation of the TTmAbs trastuzumab and pertuzumab with recombinant hyaluronidase in one ready-to-use, fixed-dose vial plus chemotherapy was approved by the U.S. Food and Drug Administration (FDA) and European Medicines Agency (EMA) for patients with HER2-positive early and metastatic breast cancer (FeDeriCa trial [58], MetaPHER trial [59]). In these open-label phase III trials, non-inferiority, safety and tolerability were satisfactorily addressed. Therefore, they paved the way to improve the patients’ quality of life by significantly reducing the treatment time for patients, physicians, nurses and pharmacy staff. This sub-cutaneous formulation brings opportunities for more flexible home management of patients with cancer.

However, not all phase III clinical trials on TTmAb combinations produced positive results. For example, in HER2-positive gastric cancer (JACOB study), the combination of pertuzumab and trastuzumab with cisplatin or 5-fluorouracil did not improve patient survival [60]. Similarly, the CAIRO2 study [61] on the hetero-combination of bevacizumab/cetuximab with oxaliplatin and capecitabine, the PACCE study [26] on bevacizumab/panitumumab combined with oxaliplatin and irinotecan in metastatic colorectal cancer, and the AVEREL [27] study on trastuzumab and bevacizumab in HER2-amplified breast cancer also were unsuccessful. Therefore, the co-targeting of VEGF and of EGFR or HER2 does not seem to be relevant in terms of synergy or additivity, possibly due to negative interactions between signaling pathways, or pharmacodynamic interactions (lack of tumor vascularization, inhibition of the expression of one of the two receptors) [62,63].

Only 20–30% of patients with metastatic melanoma responds to monotherapy with anti-CTLA-4 or anti-PD-1 antibodies to block immune checkpoints. In 2013, a phase I clinical trial in which ipilimumab (anti-CTLA-4 mAb) was combined with nivolumab (anti-PD1 mAb) reported tumor regression in 50% of treated patients [64]. In the phase III Checkmate 067 clinical trial (*n* = 945 patients with metastatic melanoma), progression-free survival was longer in the arm treated with the ipilimumab/nivolumab hetero-combination (11.5 months) than in the arms treated with nivolumab (6.9 months) or ipilimumab (2.9 months) alone [65,66]. However, this survival benefit was associated with increased toxicity in the nivolumab/ipilimumab hetero-combination arm compared with the two monotherapy arms (55% of patients with grade 3 and 4 adverse events in the hetero-combination arm vs. 16% in the nivolumab and 27% in the ipilimumab arms) [66]. This trial led the FDA to approve this hetero-combination of anti-ICM antibodies for metastatic melanoma. Since then, phase III clinical trials on anti-CTLA-4 antibodies combined with anti-PD-1 or anti-PD-L1 antibodies have shown positive results in lung cancer and renal cell carcinoma. In lung cancer, the combination of the anti-PD-L1 antibody atezolizumab with the anti-VEGF TTmAb bevacizumab, associated with chemotherapy, has shown a benefit in terms of progression-free survival compared with the arm without atezolizumab (8.3 months vs. 6.8 months) [67]. Similarly, the phase I/II trial PANACEA, which tested the anti-ICM pembrolizumab combined with the TTmAb trastuzumab in patients with HER2-amplified breast cancer, showed an improved clinical benefit in the subset of patients with PD-L1-positive tumors [68].

To date, the cocktail of nivolumab (anti-PD1 antibody) and ipilimumab (anti-CTLA-4 antibody) was the first approved and remains the only anti-ICM antibody combination approved in the clinic as first-line treatment for untreated patients with metastatic melanoma. The latest clinical data for melanoma showed up to 4 years of survival in 53% of patients receiving this hetero-combination [69]. Its use has been extended also to patients with low-risk renal carcinoma [70] and mismatch repair-deficient colorectal cancer [71]. The optimism for these mAb-based combination treatments in overcoming therapeutic resistance in different malignancies is very high. They also improve survival compared with platinum-based chemotherapy in advanced non-small cell lung cancer [72]. The combination of ipilimumab and nivolumab has been approved by the FDA for all patients with tumor displaying ≥1% of PD-L1 expression. Although these antibodies are currently used in clinical practice, some questions remain unanswered, such as the best-treatment strategy, the role of different biomarkers for patient selection and the effectiveness of immunotherapy according to specific clinical characteristics.

## 4. Antibody Combination in Phase II and Phase III Clinical Trials: A 2021 Update in Oncology

More than 300 phase II/III clinical trials in patients with cancer to test combinations of antibodies, mainly against ICMs, angiogenic factors, CD20 and receptor tyrosine kinases (RTKs), combined or not with chemotherapy, radiotherapy, small molecules or vaccines, are currently registered and in progress [15,73,74,75,76] (www.clinicaltrials.gov (accessed on 3 May 2021)). Most of these studies, completed, active or recruiting (as listed in Table 1), involve the combination of ipilimumab (anti-CTLA-4 antibody) and nivolumab (anti-PD-1 antibody) (more than 80 trials [76]), tremelimumab (anti-CTLA-4 antibody) and durvalumab (anti-PD-L1 antibody) (more than 25 trials), trastuzumab or anti-CD20 antibodies combined with anti-ICM antibodies (more than 10 and 15 trials, respectively), bevacizumab (anti-VEGF-A antibody) and atezolizumab (anti-PD-L1 antibody) (more than 10 trials) and trastuzumab combined with pertuzumab (anti-HER2 antibody) (more than 5 trials [75]). It is worth noting that most of the listed phase II/III clinical trials concern antibodies targeting ICMs (CTLA-4, PD1/PD-L1, LAG-3, TIM-3, GITR, TIGIT, CD73, ICOS, PVRIG) combined or not with TTmAbs. Conversely, fewer than 25 trials concern only TTmAb combinations.

In addition to “classical” naked antibodies, new formats, such as bispecific antibodies, probodies, antibody–drug conjugates, immunocytokines, immune-stimulating antibody conjugates or chimeric antigen receptor T (CAR-T) cells, also are included in the antibody combinations tested in phase II/III clinical trials (Table 1). Interestingly, triple antibody combinations are emerging (more than 15 trials), mainly using anti-PD1/anti-CTLA-4 antibodies with antibodies against RTKs, other ICMs (GITR, TIGIT), killer-cell immunoglobulin-like receptor, antibody–drug conjugates or immunocytokines. Triple combinations of anti-PD-L1 antibodies and TTmAbs (against RTKs, and anti-CD20 and anti-CD79b antibody-drug conjugates), or ICM inhibitors (anti-CD137, anti-CTLA-4 antibodies) also have been assessed for cancer management, as well as the anti-PD1/TIGIT/PVRIG and anti-PD1/CD40/CFS1 combinations.

Most of these antibody combinations are administered sequentially, using antibodies developed individually as active substances and initially licensed as single-agent mAbs. The treatment sequence has not been optimized yet, especially when TTmAb and anti-ICM antibody combinations are proposed, or when a chemotherapeutic agent also is added. The delivery of antibody combinations (schedule, dose and timing) has to be carefully addressed to improve the pharmacokinetics and pharmacodynamics in patients and to avoid toxicities and side effects.

## 5. Optimization of the Formulation and Delivery of Antibody Combinations

In a few cases, new industrial manufacturing/formulation strategies, developed by some pharmaceutical companies, have allowed the production of antibody combinations to be rationalized and optimized (Figure 2). Currently, antibodies for therapeutic combinations are produced and administered following four main strategies (Figure 2) that have led to the clinical development of oligoclonal antibody mixtures (Table 1 and Table 2).

### 5.1. Sequential Administration

The “antibody” active substances are produced separately (one cell line for each antibody; mainly CHO cells), the pharmaceutical formulations are made individually, and the antibody biologics are injected sequentially. This is the most classical pharmaceutical strategy when using already approved “mAb” biologics. It is the basis for the approval of the trastuzumab/pertuzumab and nivolumab/ipilimumab combinations by regulatory authorities and is used in most ongoing clinical trials in oncology (Table 1).

### 5.2. Single Co-Formulation

The “antibody” active substances are produced separately (one cell line for each antibody; mainly CHO cells), and during the pharmaceutical formulation step, the active ingredients are mixed to obtain to the “combination” biologics. The single co-formulation has been used to develop the approved combination of trastuzumab and pertuzumab (two TTmAbs) plus recombinant hyaluronidase for sub-cutaneous delivery in breast cancer, with a 1:1 or 1:2 stoichiometry depending on the formulations [58,59]. This strategy has also been employed to produce the Sym004 combination (1:1 stoichiometry) of two anti-EGFR antibodies (futuximab and modotuximab) [77,78,79] that is currently evaluated in phase II clinical trials in metastatic colorectal cancer and glioblastoma. This strategy has also been used to produce MM-151 [14,80,81], a 2:2:1 stoichiometric mixture of three anti-EGFR antibodies. This mixture has been evaluated in phase I trials in combination with chemotherapy or the anti-HER3 MM-121 antibody in colorectal and lung cancer (Table 2). In infectious diseases, a combination of three antibodies against the *Clostridium botulinum* neurotoxin has been developed and tested in a phase I trial [82]. Similarly, the single formulation of three antibodies (atoltivimab + maftivimab + odesivimab) has been approved by the FDA in 2020 for Ebola hemorrhagic fever. As demonstrated for the co-formulation of six approved antibodies [83], it seems that therapeutic monoclonal antibodies of the IgG1 subclass can be combined without severe detrimental effects to the stability of these proteins in binary mixtures.

### 5.3. Single-Cell Line Manufacturing

The active “antibody” substances are produced together in a single cell line. A single pharmaceutical formulation leads to the “combination” biologics. This Oligoclonics^®^ process [84,85] uses the PER.C6 cell line transfected with a construct that encodes a single kappa light chain, and two constructs, each encoding a heavy H chain of different specificity (“common light chain” technology). This approach has been used to produce mixtures of mono- and bi-specific antibodies. Moreover, this technology has been combined with CH3 domain engineering to force the preferential production of bispecific antibodies (Biclonics^®^). Some bispecific antibodies are currently in clinical development, such as antibodies targeting HER2/HER3 [86,87] in breast, pancreatic and gastric cancer, EGFR/leucine-rich repeat containing G protein-coupled receptor 5 (LGR5) in solid tumors, CD3/C-type lectin domain family 12 member A (CLEC12A) in acute myeloid leukemia [88], PD-L1/CD137 and EGFR/cMET in solid tumors.

### 5.4. Single Batch Manufacturing

The cell lines producing the active drug substances are initially mixed to generate a polyclonal master-cell bank. A unique pharmaceutical formulation leads to the “combination” biologics. The process allows the site-specific integration of each antibody construct on the same chromosomal locus in each cell line (Flp-In, CHO, CHO-DG44) [89,90] to standardize the expression level of each antibody after mixing the transformed cell lines. For instance, a controlled mixture of 25 anti-rhesus D antibodies (rozrolimupab or Sym001 [90]) was produced and tested in a phase II trial in patients with thrombocytopenic purpura [50]. Moreover, a mixture of two anti-cMET antibodies (1:1 stoichiometry; Sym015 [10,11,13]) was assessed in cMET-amplified tumors (phase II trial), and an oligoclonal mixture of two anti-EGFR antibodies, two anti-HER2 antibodies and two anti-HER3 antibodies (PanHER or Sym013 [12,30,31,32,91,92,93]) in epithelial cancers (phase I trials).

## 6. Challenges and Regulation of Antibody Combinations

Homo- and hetero-combinations of antibodies have many advantages compared with antibody monotherapy (Table 3). Antibody combinations allow several well-defined epitopes on one or more antigens to be targeted with a perfectly controlled and adjustable antibody stoichiometric ratio. Within an antibody cocktail, the affinity, epitope, isotype, or glycosylation of each antibody can be tailored.

Antibodies are glycoproteins that contain a glycosylation site at position 297 in the Fc region. TTmAbs need to be N-glycosylated to display effector functions. Moreover, the presence or absence of terminal sugars on the glycans in the Fc region strongly influence the antibody pharmacokinetics (e.g., high mannose content decreases the antibody half-life), pharmacodynamics, stability, safety (immunogenicity, specifically when the antibody is derived from non-human cells) and efficacy. The effector functions (ADCC and complement-dependent cytotoxicity) can also be affected. Therefore, glycoforms should be thoroughly analyzed in each antibody batch produced to offer stable and safe antibodies, an essential step for their successful clinical translation. Glycoengineering strategies have been developed to produce antibodies harboring the desired glycoforms in order to enhance their efficacy and safety [94,95].

Targeting several epitopes on the same receptor or on several receptors allows the number of targets recruited to be increased and thus an increased number of antibodies bound per cell. The biological responses induced by these antibodies are enhanced, such as the Fc-dependent immune effector mechanisms of IgGs (ADCC, complement-dependent cytotoxicity, antibody-dependent cell phagocytosis), and the inhibition of compensatory cell signaling (possibly through the target elimination or internalization/degradation) based on the Fab portion of IgGs. In hetero-combinations of antibodies against immune checkpoints or their ligands (e.g., PD-L1), the combination simultaneously counteracts the redundant negative regulatory mechanisms exploited by tumors to escape the immune system. Therefore, an oligoclonal mixture of antibodies allows their spectrum of activity to be broadened by anticipating and avoiding possible resistance to the treatment (pre-existing or acquired through the emergence of resistant clones under the effect of one of the combination components). Finally, in oncology, hetero-combinations to target cancer cells and/or the tumor immune microenvironment and vessels might implicate synergistic mechanisms of action in vivo, the modalities of which remain to be clarified. In addition, antibody combinations could also pose unique intellectual property challenges [96].

The production of antibody mixtures is generally regulated by the good manufacturing practices conventionally used to produce single therapeutic antibodies [97]. Currently, the strategy of “sequential administration” is still predominant for combinations of two mAbs, in approved formulations and also in clinical trials. However, if mixtures of three to six antibodies are going to be developed and tested, the cost of this strategy will progressively increase, and regulatory procedures and constraints will become more cumbersome and complicated. Indeed, according to the EMA and FDA regulations, the toxicity, efficacy and pharmacokinetics of each component of an oligoclonal mixture must be evaluated individually and in combination. Therefore, the regulatory procedures are duplicated for each active substance, because each antibody is considered as a single biologic in the combination. Faced with these difficulties, the “single co-formulation”, or the “single cell” and “single batch” manufacturing should allow the production processes and costs to be controlled and the regulatory and registration procedures to be simplified [98]. Concretely, the FDA has already authorized oligoclonal antibody mixtures, prepared according to the “single co-formulation” strategy, for use in phase I and II clinical trials [82]. Moreover, the FDA has approved phase I trials with Sym004 [99] and MM-151 [100], obtained using the “single co-formulation” strategy, to be tested as a “single biologic” in oncology. The combination of trastuzumab and pertuzumab plus hyaluronidase, produced by single co-formulation, has been approved by the FDA and EMA. Therefore, it is necessary to choose, already at the preclinical stage, the production strategy of oligoclonal mixtures to reduce risks and costs. It remains to be seen how the FDA and EMA will consider, from a regulatory point of view, the new oligoclonal antibody mixtures prepared using “single cell” and “single batch” manufacturing.

Compared with single-agent therapies, determining the optimal dose and schedule of each mAb is crucial in combination regimens. An oligoclonal mixture is formulated according to a stoichiometric ratio of antibodies defined at the time of the initial regulatory application for an investigational new drug. Due to the specific pharmacokinetics of each antibody (absorption, distribution, metabolism and excretion), the initial stoichiometric formulation is unlikely to be maintained in the patient during treatment. This problem must be addressed in preclinical studies. The dose and treatment sequence choice should take into account the specific pharmacokinetic features of each antibody in the mixture. For example, the stoichiometric ratio of the six antibodies in the Sym013 mixture varies in vivo over time in function of target exposure, ranging from high for EGFR to medium or low for HER2 and HER3. The mechanisms of action of the oligoclonal mixture, compared with each antibody in the mixture, must be determined in relevant preclinical models to support clinical development.

The challenge today is to better understand signaling networks with the ultimate aim of developing combination regimens or adaptive sequential strategies that translate high partial response rates to durable complete responses. Clinical observations highlight the importance of flexible approaches to optimize the dose and schedule of mAb-based combinations [101]. The toxicity potentiation observed in some combination trials [23] underscores the need of careful dose titration in phase I clinical trials. The two clinically approved combinations use the antibody doses identified in the monotherapy clinical trials. However, the dose chosen in phase I clinical trials of new antibody combinations may not necessarily be the same as the dose approved for monotherapy. For instance, a recent follow-on study evaluated the combination of vemurafenib and ipilimumab using a sequential schedule of administration [102]. This regimen demonstrated a substantially improved safety profile, with marked reduction of hepatotoxicity compared with the previous study in which ipilimumab and vemurafenib were administered concurrently. This study clearly highlights the clinical development challenges and risks of combining anti-cancer antibodies at standard doses and schedules. In addition, it is clear that the optimal dose and schedule for a given combination may differ in function from the indication due to differences in disease biology and/or co-morbidities in the various patient populations. Although pre-clinical animal models have limitations, they can be useful to assess the therapeutic potential of specific combinations by unraveling their mechanisms of action and providing insights into the underlying biology of various therapeutic strategies. Moreover, in the context of cancer therapies, mathematical and computational approaches are becoming more and more relevant to overcome the various challenges related to the optimization of combined protocols to obtain synergistic effects [103,104,105].

## 7. Conclusions: Towards a Therapeutic Polyclonal Immune Response?

The immune system has naturally evolved towards a polyclonal humoral response. The development of oligoclonal antibody combinations and mixtures to improve the existing targeted therapies is approaching the goal of mimicking the natural humoral immune response. However, practical and regulatory biological constraints must be overcome to enrich this pharmacological class of antibodies.

## Figures and Tables

**Figure 1 cancers-13-04620-f001:**
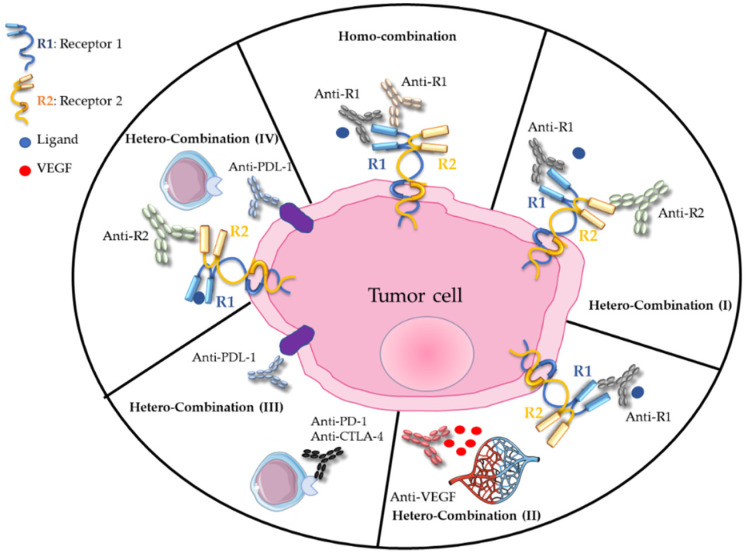
Homo- and hetero-combinations of monoclonal antibodies, adapted from [15].

**Figure 2 cancers-13-04620-f002:**
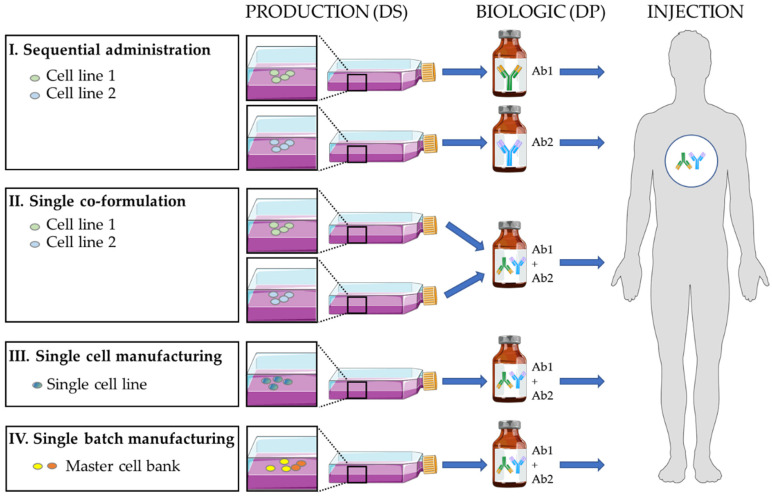
Different strategies to produce a biologic as a drug product (DP) that is a mixture of two antibody drug substances (DS). The single-cell manufacturing allows a mixture of mono- and bi-specific antibodies to be produced. Ab: antibody.

**Table 1 cancers-13-04620-t001:** Antibody-based drug combinations currently examined by clinical trials at the date of 3 May 2021, updated from [73].

Drug Combination	Antibody Targets	Major Tumor Types	ClinicalTrial.gov Identifier (NTC0)
Anti-PD-1 + ADG106	PD-1 + CD137	Solid cancers, NHL	4775680
Anti-PD-1 + LYT-200	PD-1 + galectin 9	Solid cancers	4666688
Anti-PD-1 + TJ004309	PD-1 + CD73	Solid cancers	4322006
Anti-PD-L1 + IMC-F106C	PD-L1 + CD3/PRAME (BsAb)	PRAME-cancers	4262466
Atezolizumab + ado-trastuzumab emtansine	PD-L1 + HER2 (ADC)	Breast cancer	2924883
Atezolizumab + anetumab ravtansine	PD-L1 + mesothelin (ADC)	NSCLC	3455556
Atezolizumab + daratumumab	PD-L1 + CD38	NSCLC	3023423
Atezolizumab + isatixumab	PD-L1 + CD38	Solid cancers	3637764
Atezolizumab + KY1044	PD-L1 + ICOS	Solid cancers	3829501
Atezolizumab + mosunetuzumab	PD-L1 + CD20/CD3 (BsAb)	NHL, CLL	2500407
Atezolizumab + obinutuzumab + CT	PD-L1 + CD20	FL	2596971
Atezolizumab + obinutuzumab + ibrutinib	PD-L1 + CD20	CLL	2846623
Atezolizumab + obinutuzumab + lenalidomide	PD-L1 + CD20	FL	2631577
Atezolizumab + obinutuzumab + polatuzumab vedo, Atezolizumab + rituximab + polatuzumab vedo	PD-L1 + CD20 + CD79b (ADC), PD-L1 + CD20 + CD79b (ADC)	FL, DLBCL	2729896
Atezolizumab + pertuzumab + trastuzumab	PDL-1 + HER2 + HER2	Breast cancer	3417544
Atezolizumab + RO6958688	PD-L1 + CEA/CD3 (BsAb)	NSCLC	3337698
Atezolizumab + tiragolumab Atezolizumab + tiragolumab + CT	PD-L1 + TIGIT PD-L1 + TIGIT	NSCLC Esophageal cancer	3563716, 4294810 4540211
Atezolizumab + trastuzumab + CT	PD-L1 + HER2	GC, GEJ	4661150
Atezolizumab + trastuzumab + pertuzumab + CT	PD-L1 + HER2 + HER2	Breast cancer	3125928
Atezolizumab + tocilizumab	PD-L1 + IL-6R	Prostate cancer	3821246
Atezolizumab + tocilizumab + RT	PD-L1 + IL-6R	Astrocytoma, GBM	4729959
Avelumab + ivuxolimab, ivuxolimab + utomilumab, Avelumab + utomilumab	PD-L1 + OX40, OX40 + CD137, PD-L1 + CD137	AML, breast cancer	3390296, 3971409
Avelumab + utomilumab + rituximab	PD-L1 + CD137 + CD20	DLBCL	2951156
Avelumab + utomilumab, avelumab + PD-0360324, Avelumab + utomilumab + ivuxolimab	PD-L1 + CD137, PD-L1 + CSF1, PD-L1 + OX40	Solid cancers	2554812
Avelumab + utomilumab, Avelumab + ivuxolimab, Avelumab + utomilumab + ivuxolimab	PD-L1 + CD137, PD-L1 + OX40, PDL1 + CD137 + OX40	Solidcancers	3217747
Balstilimab + AGEN1181	PD-1 + CTLA-4	Solid cancers	3860272
Balstilimab + zalifrelimab, Balstilimab + zalifrelimab + CT	PD-1 + CTLA-4	Angiosarcoma, cervical cancer, bladder cancer	More than 3 trials
BCD-217 + BCD-100	CTLA-4 + PD-1 + PD-1	Melanoma	3913923
Bevacizumab + AK104 + CT	VEGF-A + PD-1/CTLA-4 (BsAb)	Cervical cancer	4868708
Bevacizumab + atezolizumab	VEGF-A + PD-L1	Solid cancers	More than 5 trials
Bevacizumab + atezolizumab + CT	VEGF-A + PD-L1	CRC, NSCLC, Breast cancer	More than 3 trials
Bevacizumab + atezolizumab + eganelisib	VEGF-A + PD-L1	Breast cancer	3961698
Bevacizumab + atezolizumab + endocrine ther.	VEGF-A + PD-L1	Breast cancer	3280563
Bevacizumab + atezolizumab + entinostat	VEGF-A + PD-L1	RCC	3024437
Bevacizumab + atezolizumab + ipatasertib	VEGF-A + PD-L1	Breast, ovarian cancer	3395899, 3363867
Bevacizumab + atezolizumab, Atezolizumab + ladiratuzumab vedotin	VEGF-A + PD-L1, PD-L1 + LIV-1 (ADC)	Breast cancer	3424005
Bevacizumab + avelumab + Ad-CEA vax + CT	VEGF-A + PD-L1	CRC	3050814
Bevacizumab + BCD-100 + CT	VEGF-A + PD-1	Cervical cancer	3912402, 3912415
Bevacizumab + brentuximab vedotin	VEGF-A + CD30 (ADC)	Germ cell tumor	2988843
Bevacizumab + camrelizumab	VEGF-A + PD-1	GTD	4812002
Bevacizumab + carotuximab	VEGF-A + endoglin	GTD	2664961
Bevacizumab + cetuximab + CT	VEGF-A + EGFR	CRC	0265850
Bevacizumab + durvalumab + CT	VEGF-A + PD-L1	Ovarian cancer	3737643
Bevacizumab + pembrolizumab	VEGF-A + PD-1	RCC	2348008
BI-1206 + CD20 Ab	CD32b + CD20	BCL	2933320
Cemiplimab + isatuximab	PD-1 + CD38	MM, lymphoma	3194867, 3769181
Cemiplimab + REGN5668, Ubamatamab + REGN5668	PD-1 + MUC16/CD28 (BsAb) MUC16/CD3 + MUC16/CD28	Ovarian cancer	4590326 3564340
Cemiplimab + REGN7075	PD-1 + EGFR/CD28 (BsAb)	Solid cancers	4626635
Cemiplimab + SAR439459	PD-1 + TGFβ	Solid cancers	3192345
Cetrelimab + daratumumab	PD-1 + CD38	Solid cancers	3547037
Cetuximab + avelumab	EGFR + PD-L1	HNSCC	3494322
Cetuximab + Hu5F9-G4	EGFR + CD47	CRC	2953782
CX-2009 + CX-072	CD166 (PDC) + PD-1 (PDC)	Breast cancer	4596150
Dostarlimab + cobolimab	PD-1 + TIM3	HCC	3680508
Durvalumab + axatilimab	PD-L1 + CSF1-R	CC	4301778
Durvalumab + daratumumab	PD-L1 + CD38	MM	2807454
Durvalumab + cetuximab + RT	PD-L1 + EGFR	HNSCC	3051906
Durvalumab + monalizumab	PD-L1 + NKG2A	NSCLC	3822351, 3794544
Durvalumab + oleclumab, Durvalumab + oleclumab + RT	PD-L1 + CD73	Breast cancer, NSCLC	3875573, 3822351, 3794544
Durvalumab + rituximab	PD-L1 + CD20	Lymphoma, CLL	2733042
Enoblituzumab + retifenlimab, Retifenlimab + MGC018 Enoblituzumab + tebotelimab	B7-H3 + PD-1, PD-1 + B7-H3 (ADC) B7-H3 + PD-1/LAG-3 (DART)	HNC, solid cancers	4633485, 3729596
Iodine-131 tositumomab + rituximab + CT	CD20 + CD20	NHL	0770224
Ipilimumab + cemiplimab + CT	CTLA-4 + PD-1	NSCLC	3409614, 3430063
Ipilimumab + envafolimab	CTLA-4 + PD-L1	Sarcoma	4480502
Ipilimumab + nivolumab	CTLA-4 + PD-1	Solid/hematological cancers	More than 60 trials
Ipilimumab + nivolumab + CT	CTLA-4 + PD-1	Sarcoma, NSCLC, TCC	3138161, 3215706, 3036098
Ipilimumab + nivolumab + DC-based vaccine	CTLA-4 + PD-1	SCLC, RCC	3406715, 4203901
Ipilimumab + nivolumab + epacadostat, Nivolumab + lirilumab + epacadostat	CTLA-4 + PD-1, PD-1 + KIR	Solid cancers	3347123
Ipilimumab + nivolumab + TKIs	CTLA-4 + PD-1	Melanoma	4655157
Ipilimumab + nivolumab + glembatumumab vedotin	CTLA-4 + PD-1 + GPNMB (ADC)	Solid cancers	3326258
Ipilimumab + nivolumab + ragilifimab, Ragilifimab + ipilimumab, Ragilifimab + nivolumab	CTLA-4 + PD-1 + GITR, GITR + CTLA-4, GITR + PD-1	Solid cancers	3126110
Ipilimumab + nivolumab + lirilumab, Nivolumab + lirilumab	CTLA-4 + PD-1 + KIR, PD-1 + KIR	Solid cancers	1714739
Ipilimumab + nivolumab + panitumumab	CTLA-4 + PD-1 + EGFR	CRC	3442569
Ipilimumab + nivolumab + prednisolone	CTLA-4 + PD-1	Melanoma	3563729
Ipilimumab + nivolumab + RT	CTLA-4 + PD-1	NSCLC, PDAC, OSCC	More than 3 trials
Ipilimumab + nivolumab + IT-hu14,18-IL2 +RT	CTLA-4 + PD-1 + GD2-IL2 (IC)	Melanoma	3958383
Ipilimumab + nivolumab + TKIs	CTLA-4 + PD-1	CRC, NSCLC	More than 3 trials
Ipilimumab + nivolumab + trastuzumab, Nivolumab + trastuzumab + CT	CTLA-4 + PD-1 + HER2, PD-1 + HER2	GC	3409848
Ipilimumab + nivolumab, nivolumab + BMS-986016, Nivolumab + daratumumab	CTLA-4 + PD-1, PD-1 + LAG3, PD-1 + CD38	CRC	2060188
Ipilimumab + nivolumab, nivolumab + lirilumab, Nivolumab + daratumumab	CTLA-4 + PD-1, PD-1 + KIR, PD-1 + CD38	Hematological cancers	1592370
Ipilimumab + pembrolizumab	CTLA-4 + PD-1	Melanoma	2743819
Ipilimumab + vopratelimab	CTLA-4 + ICOS	NSCLC, UC	3989362
Magrolimab + mogamulizumab	CD47 + CCR4	T-cell lymphoma	4541017
MGD007 + retifanlimab	gpA33/CD3 (DART) + PD-1	CRC	3531632
Margetuximab + retifanlimab +/− CT Margetuximab + tebotelimab + CT	HER2 + PD-1 HER2 + PD-1/LAG-3 (DART)	GC, GEJ	4082364
Nivolumab + andecaliximab	PD1 + MMP9	GC, GEJ	2864381
Nivolumab + anetumab ravtansine, Nivolumab + ipilimumab + anetumab ravtansine +/− CT	PD-1 + mesothelin (ADC), PD-1 + mesothelin (ADC) + CTLA-4	PDAC	3816358
Nivolumab + bevacizumab Nivolumab + bevacizumab + RT	PD-1 + VEGF-A	Ovarian, peritoneal cancer, GBM	2873962 3743662
Nivolumab + BA3011	PD-1 + AXL (CAB-ADC)	NSCLC	4681131
Nivolumab + blinatumomab	PD-1 + CD3/CD19 (BsAb)	B-ALL	4546399
Nivolumab + BMS-986012 +/− CT	PD-1 + fucosyl-GM1	SCLC	2247349, 4702880
Nivolumab + BMS-986179	PD-1 + CD73	Solid cancers	2754141
Nivolumab + BMS-986207 + COM701	PD-1 + TIGIT + PVRIG	Solid cancers	4570839
Nivolumab + BMS-986218	PD-1 + CTLA-4	Solid cancers	3110107, 4785287
Nivolumab + BMS-986249	PD-1 + CTLA-4 (PDC)	Solid cancers	3369223
Nivolumab + BMS-986253	PD-1 + IL-8	Solid cancers	3400332, 3689699
Nivolumab + brentuximab vedotin	PD-1 + CD30 (ADC)	HL, NHL	2572167, 2581631
Nivolumab + elotuzumab	PD-1 + SLAMF7	MM	2612779, 3227432
Nivolumab + etigilimab	PD-1 + TIGIT	Solid cancers	4761198
Nivolumab + nimotuzumab	PD-1 + EGFR	NSCLC	2947386
Nivolumab + oregovomab	PD-1 + CA125	Ovarian cancer	3100006
Nivolumab + relatlimab	PD-1 + LAG-3	Solid cancers, melanoma, HNSCC, CRC	More than 3 trials
Nivolumab + rituximab + CT	PD-1 + CD20	DLBCL	3259529
Nivolumab + rituximab + lenalidomide	PD-1 + CD20	DLBCL, CNS lymphoma	3558750
Nivolumab + sotigalimab	PD-1 + CD40	Melanoma, NSCLC	3123783
Nivolumab + sotigalimab + cabiralizumab	PD-1 + CD40 + CSF1R	Melanoma, NSCLC, RCC	3502330
Nivolumab + urelumab	PD-1 + CD137	Solid cancers, NHL, TCC	2253992
Nivolumab + varlilumab	PD-1 + CD27	Solid cancers, BCL	2335918, 3038672
Nivolumab and/or ipilimumab + BMS-986178	PD-1 and/or CTLA-4 + OX40	Solid cancers	2737475
Nivolumab or pembrolizumab + glembatumumab vedotin, Glembatumumab vedotin + varlilumab	PD-1 + GPNMB (ADC), GPNMB (ADC) + CD27	Melanoma	2302339
Obinutuzumab + glofitamab + CD19-CAR-T Obinutuzumab + glofitamab	CD20 + CD20/CD3 (BsAb)+CD19, CD20 + CD20/CD3 (BsAb)	DLBCL, Lymphomas	4889716 4703686
Obinutuzumab + polatuzumab vedotin	CD20 + CD79b (ADC)	NHL	1691898
Obinutuzumab + polatuzumab vedotin + CT, Rituximab + polatuzumab vedotin + CT	CD20 + CD79b (ADC), CD20 + CD79b (ADC)	FL, DLBCL	2600897, 2611323
Pembrolizumab + anetumab ravtansine	PD-1 + mesothelin (ADC)	Mesothelioma	3126630
Pembrolizumab + bavituximab	PD-1 + phosphatidylserine	HCC	3519997
Pembrolizumab + BDC-1001	PD-1 + HER2 (ISAC)	HER2+ cancers	4278144
Pembrolizumab + BI-1206	PD-1 + CD32b	Solid cancers	4219254
Pembrolizumab + BI-1808	PD-1 + TNFR2	Solid cancers	4752826
Pembrolizumab + brentuximab vedotin	PD-1 + CD30 (ADC)	T-cell lymphoma	4795869
Pembrolizumab + canakinumab + CT	PD-1 + IL-1b	NSCLC	3631199
Pembrolizumab + cetuximab Pembrolizumab + trastuzumab, Pembrolizumab + ado-trastuzumab emtansine	PD-1 + EGFR PD-1 + HER2, PD-1 + HER2 (ADC)	Solid cancers, HNSCC	2318901, 3082534
Pembrolizumab + feladilimab +/− CT	PD-1 + ICOS	HNSCC	4428333, 4128696
Pembrolizumab + mirvetuximab soravtansine	PD-1 + FRα (ADC)	Endometrial cancer	3835819
Pembrolizumab + mogamulizumab	PD-1 + CCR4	Lymphoma	3309878
Pembrolizumab + NP137	PD-1 + Netrin-1	Gynecological cancer	4652076
Pembrolizumab + quavonlimab, Pembrolizumab + vibostolimab, Pembro + quavonlimab + vibostolimab	PD-1 + CTLA-4, PD-1 + TIGIT, PD-1 + CTLA-4 + TIGIT	Melanoma	4305054, 4305041, 4303169
Pembrolizumab + sotigalimab	PD-1 + CD40	Melanoma	2706353
Pembrolizumab + vilobelimab	PD-1 + C5a	SCC	4812535
Pembrolizumab + vofatamab	PD-1 + FGFR3	TCC	3123055
Rituximab + belimumab	CD20 + BAFF	CSC	3844061
Rituximab + BI-1206	CD20 + CD32b	NHL	3571568
Rituximab + Hu5F9-G4	CD20 + CD47	NHL	2953509
Rituximab + ibritumomab tiuxetan	CD20 + CD20 (ARC)	NHL	732498
Rituximab + ibritumomab tiuxetan + CT	CD20 + CD20 (ARC)	FL, NHL	372905
Serplulimab + HLX04 +/− CT	PD-1 + VEGF-A	CRC, HCC, NSCLC	More than 3 trials
Serplulimab + HLX07	PD-1 + EGFR	HNC	4297995
Sintilimab + camrelizumab +/− apatinib +/− CT	PD-1 + PD-1	Solid cancers	4282278
Sintilimab + IBI305	PD-1 + VEGF-A	HCC	3794440
Sintilimab + IBI310	PD-1 + CTLA-4	Cervical cancer, CRC	4590599, 4258111
Spartalizumab + lacnotuzumab	PD-1 + CSF1	ESCC	3785496
Spartalizumab + LAG525	PD-1 + LAG-3	Solid and hematological cancers, breast cancer	3499899, 2460224, 3365791
Spartalizumab + MBG454	PD-1 + TIM-3	Solid cancers	2608268
Spartalizumab + NIS793 +/− CT	PD-1 + TGF-β	Solid cancers, PDAC	4390763, 2947165
Tislelizumab + BGB-A425	PD-1 + TIM-3	Solid cancers	3744468
Tislelizumab + garivulimab	PD-1 + PD-L1	Solid cancers	3379259, 4702009
Tislelizumab + ociperlimab	PD-1 + TIGIT	Lung cancer, ESCC	4746924, 4732494, 4693234
Tislelizumab + zanidatamab + CT	PD-1 + HER2/HER2 (BsAb)	Breast, GC, GEJ	4276493
Tocilizumab + CC-1	IL-6R + PSMA/CD3 (BsAb)	SCC	4496674
Toripalimab + YH003	PD-1 + CD40	Solid cancers	4481009
Trastuzumab + avelumab + CT, Trastuzumab + avelumab + utomilumab +/− CT	HER2 + PD-L1, HER2 + PD-L1 + CD137	Breast cancer	3414658
Trastuzumab + camrelizumab + CT	HER2 + PD-1	GC, GEJ	3950271
Trastuzumab + envafolimab	HER2 + PD-L1	Breast cancer	4043195
Trastuzumab + necitumumab + osimertinib	HER2 + HER2	NSCLC	4285671
Trastuzumab + pembrolizumab + CT	HER2 + PD-1	GC	2901301
Trastuzumab + pertuzumab	HER2 + HER2	Breast cancer	2625441
Trastuzumab + pertuzumab + CT	HER2 + HER2	Breast cancer	1796197, 2402712
Trastuzumab + pertuzumab + copanlisib	HER2 + HER2	Breast cancer	4108858
Trastuzumab + pertuzumab + durvalumab	HER2 + HER2 + PD-L1	Breast cancer	3820141
Trastuzumab + QL1209 + CT	HER2 + HER2	Breast cancer	4629846
Trastuzumab deruxtecan + durvalumab + CT, Trastuzumab deruxtecan + pertuzumab	HER2 (ADC) + PD-L1, HER2 (ADC) + HER2	Breast cancer	4538742
Trastuzumab + zenocutuzumab +/− CT	HER2 + HER2/HER3 (BsAb)	Breast cancer	3321981
Tremelimumab + durvalumab	CTLA-4 + PD-L1	Solid cancers	More than 25 trials
Tremelimumab + durvalumab + CT	CTLA-4 + PD-L1	HNSCC, CRC, SCLC	3019003, 3202758, 3043872
Tremelimumab + durvalumab + hormone	CTLA-4 + PD-L1	Breast cancer	3430466
Tremelimumab + durvalumab + IMCgp100	CTLA-4 + PD-L1	Melanoma	2535078
Tremelimumab + durvalumab + olaparib	CTLA-4 + PD-L1	Ovarian, peritoneal cancer	2953457
Tremelimumab + durvalumab + proton therapy	CTLA-4 + PD-L1	HNSCC	3450967
Tremelimumab + durvalumab + RT	CTLA-4 + PD-L1	PDAC, HNSCC, HCC, CRC	More than 3 trials
Tremelimumab + feladilimab	CTLA-4 + ICOS	Solid cancers	3693612
Tripleitriumab + QL1101	PD-1 + VEGF-A	CRC	4527068
Zimberelimab + Domvanalimab	PD-1 + TIGIT	NSCLC, SCC, Lung cancer	4736173, 4262856

This table lists all current phase II and III clinical studies (not yet recruiting or recruiting, enrolling, active, completed or terminated) testing two or more antibodies in patients with cancer (phase I studies were excluded). The information was extracted from www.clinicaltrials.gov (accessed on 3 May 2021). Only the major clinical indications are listed. The clinical trial identifiers have been abbreviated. ACC, adenoid cystic carcinoma, ADC, antibody–drug conjugate; Ad-CEA, carcinoembryonic antigen; AML, acute myeloid leukemia; BCL, B-cell lymphoma; BiTE, bispecific T-cell engager; BTC, biliary tract cancer; CC: cholangiocarcinoma; CLL, chronic lymphocytic leukemia; CNS, central nervous system; CRC, colorectal cancer; CSC, cutaneous systemic sclerosis; CT, chemotherapy; DART, dual-affinity re-targeting antibody; DC, dendritic cell; DLBCL, diffuse large B-cell lymphoma; ESCC, esophageal squamous cell carcinoma; SCC, squamous cell carcinoma; FL, follicular lymphoma; GBM, glioblastoma; GC, gastric cancer, GEJ, gastroesophageal junction cancer; GTD, gestational trophoblastic disease; HCC, hepatocellular carcinoma; HL, Hodgkin lymphoma; HNSCC, head and neck squamous cell carcinoma; IC, immunocytokine; ISAC, immune stimulating antibody conjugate; BCC, basal cell carcinoma; PDAC, pancreatic adenocarcinoma; MCL, mantle cell lymphoma; MM, multiple myeloma; NHL, non-Hodgkin lymphoma; NSCLC, non-small cell lung cancer; PDAC, pancreatic ductal adenocarcinoma; PDC, probody–drug conjugate; Polatuzumab vedo, polatuzumab vedotin; RCC, renal cell carcinoma; RMC, renal medullary carcinoma; RT, radiotherapy; SCLC, small cell lung cancer; TCC, transitional cell carcinoma; TKI, tyrosine kinase inhibitor; UC, urothelial cancer.

**Table 2 cancers-13-04620-t002:** Antibody mixtures in oncology.

Clinical Indication	Antibody	Formulation	Target	Clinical Trial	Date
Colorectal cancer/lung cancer	MM-151	Single co-formulation	3 × EGFR	Phase I *	2015
Colorectal cancer/lung cancer	MM-151 + MM-121	Sequential administration	3 × EGFR + HER3	Phase I *	2015
Colorectal cancer/glioblastoma	Sym004: futuximab + modotuximab	Single co-formulation	2 × EGFR	Phase II	2018
Epithelial cancers	Sym013	Single batch	2 × EGFR + 2 × HER2 + 2 × HER3	Phase II *	2016
c-MET amplified tumors	Sym015	Single batch	2 × cMET	Phase II	2016
Breast cancer	Trastuzumab + Pertuzumab +Hyal **	Single co-formulation	2 × HER2	Approval	2020

* Discontinued; ** Hyal: hyaluronidase.

**Table 3 cancers-13-04620-t003:** Challenges of antibody combinations.

Benefits	Questions
Multi-epitope targeting	Manufacturing of antibody mixtures
Biological synergistic effects	Cost
Aggregation	Regulatory affairs
Activation/inhibition of immune responses	Toxicities
Activation/inhibition of signaling	Different PKC for each antibody drug substance *
Target elimination	Stoichiometry of antibody drug substances
Avoiding drug resistance	Dose, timing and treatment schedule

* PKC: pharmacokinetics.

## Data Availability

No new data were created or analyzed in this study. Data sharing is not applicable to this article.

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
