# Peer review of "Improving Biologics’ Effectiveness in Clinical Oncology: From the Combination of Two Monoclonal Antibodies to Oligoclonal Antibody Mixtures"

_cancers, 2021, doi:10.3390/cancers13184620_

Round 1

Reviewer 1 Report

The paper by Larbouret aims to review current clinical efforts into combination therapy of different two or more monoclonal antibodies. There are four sub-sections that extensively describe different aspects of this topic, with comprehensive examples from past and on-going clinical trials. The first section describes homo-combinations and hetero-combinations of antibodies in preclinical studies, with specific examples of targeted-mAbs, as well as checkpoint inhibitors, and the different combinations explored in this arena. Next, the authors describe clinical trials examples as proof of concept for the use of such combined therapies in human patients. This is then followed by an updated detailed listing of current phase II/III clinical trials, with some specific examples and their outcomes. Subsequently, there is a description of the different strategies used by the industry to manufacture/design such combination therapies, also providing some insights into the challenges that need to be addressed in terms of the regulation of such combination therapies. The manuscript is well-written, easy to follow, comprehensive, and the figures and tables are highly informative. Some of the figures and tables were reproduced/modified from other publications but there was no information regarding permissions of such re-use, so this needs to be verified. Although glycosylation is mentioned briefly, perhaps it would be useful to add a short paragraph about the challenges that relate to the glycosylation on such biotherapeutics and their immunological and clinical effects, as those are also increasingly addressed by the regulatory agencies.

Author Response

Point 1: Some of the figures and tables were reproduced/modified from other publications but there was no information regarding permissions of such re-use, so this needs to be verified.

The figure 1 was modified from an open journal and we cited the authors and journal. For the Figure 2, the figure was modified and we have the permission of modification of the figure by the first author (Soren Rasmussen). For the table 1, the data are updated and so a part is completely new. The journal and author from the publication are cited in the manuscript.

Point 2 : Although glycosylation is mentioned briefly, perhaps it would be useful to add a short paragraph about the challenges that relate to the glycosylation on such biotherapeutics and their immunological and clinical effects, as those are also increasingly addressed by the regulatory agencies.

As requested, we added a short paragraph about antibody glycosylation (line 390-400).

Reviewer 2 Report

The article by Larbouret et al. is an extensive and informative summary of the current state of antibody treatments in clinical oncology. I believe, after small stylistic revisions, the manuscript will serve as a thoughtful review for those looking to learn more on the subject.

My broader comments are these:

  • Sentences are often quite long and include thoughts on several ideas simultaneously. These make it somewhat difficult to follow, and I would suggest editing the language throughout the review to avoid run-on, overly complex sentences.    
  • Similarly, the headings for most of the sections can be too long and do not serve to prepare the reader well for what the upcoming section will contain. Example of a succinct heading: section 6. Example of an overly detailed heading: section 5 (and others). Please, consider re-writing those.
  • The introductory section is informative, but the inclusion of a short overview of what the article will focus on specifically may be beneficial.

Specific concerns or comments:

  • Please define the terms “homo-combinations” and “hetero-combinations” within the first one or two sentences of section 2.1. The former is addressed (on lines 59-60 I think), but the latter is never explicitly defined.
  • On line 74, the authors refer to a “synergic effect.” Was this a typo for “synergistic” or “synergic” was chosen purposefully and perhaps a better adjective can be used.
  • The authors use the term synergistic quite a few times throughout the manuscript. Synergy is often a very specific result based on statistical outcomes; was this the outcome referred to by the authors or were the effects discussed better described as “improved” or “strengthened?” Example: use of synergy on line. 187
  • Please, refer to T-cells consistently as either T lymphocytes or T-cells (both are used throughout the text).
  • The opening sentence in section 2.3 ends with a broad statement that needs a citation.
  • The sentence beginning with “Other cocktails…” on line 255 is overly broad. Please remove or edit.
  • On several occasions the authors refer to “the association of [Ab1] and [Ab2]…” This wording was confusing; did the authors mean “the combination of [Ab1] and [Ab2]”?
  • The sentence on lines 397-398 is repeated almost verbatim on lines 438-439.
  • Please edit Figure 2 to more clearly indicate that in the “single batch manufacturing” scenario, a mixed culture of producer cell lines is used (perhaps make the oval shapes different colors). As is, it is almost identical with the schematic above.   
  • The black-on-blue format of the table headings makes them quite difficult to read, both in print and on screen.

Author Response

The article by Larbouret et al. is an extensive and informative summary of the current state of antibody treatments in clinical oncology. I believe, after small stylistic revisions, the manuscript will serve as a thoughtful review for those looking to learn more on the subject.

My broader comments are these:

  • Sentences are often quite long and include thoughts on several ideas simultaneously. These make it somewhat difficult to follow, and I would suggest editing the language throughout the review to avoid run-on, overly complex sentences.

Sentences were simplified and the manuscript corrected by a native English-speaking professional.

  • Similarly, the headings for most of the sections can be too long and do not serve to prepare the reader well for what the upcoming section will contain. Example of a succinct heading: section 6. Example of an overly detailed heading: section 5 (and others). Please, consider re-writing those.

As requested, the manuscript was modified.

  • The introductory section is informative, but the inclusion of a short overview of what the article will focus on specifically may be beneficial.

As requested, an overview of what the article will discuss was added in the Introduction (line 49-60).

Specific concerns or comments:

  • Please define the terms “homo-combinations” and “hetero-combinations” within the first one or two sentences of section 2.1. The former is addressed (on lines 54-60 I think), but the latter is never explicitly defined.

We defined the terms “homo-combinations” and “hetero-combinations” at the end of the Introduction section (line 51-52).

  • On line 74, the authors refer to a “synergic effect.” Was this a typo for “synergistic” or “synergic” was chosen purposefully and perhaps a better adjective can be used.

The term “synergic” was replaced by “improved” because no statistical analysis was performed in the quoted study.

  • The authors use the term synergistic quite a few times throughout the manuscript. Synergy is often a very specific result based on statistical outcomes; was this the outcome referred to by the authors or were the effects discussed better described as “improved” or “strengthened?” Example: use of synergy on line. 187

We agree with reviewers that the term synergy is based on specific statistical criteria.

-In paragraph 2.2, synergy has been replaced by  increase efficacy.

-In paragraph 2.3 act in synergy has been removed.

-In paragraph 3 “Synergy” has been replaced by “an improved clinical benefit”

  • Please, refer to T-cells consistently as either T lymphocytes or T-cells (both are used throughout the text).

We used T-cells throughout the text.

  • The opening sentence in section 2.3 ends with a broad statement that needs a citation.

As requested, we added a reference at the end of the opening sentence in section 2.3: Delou JMA et al. Highlight in resistance mechanisms pathways for combination therapy. Cells 2019.

  • The sentence beginning with “Other cocktails…” on line 255 is overly broad. Please remove or edit.

We removed the sentence, line 255.

  • On several occasions the authors refer to “the association of [Ab1] and [Ab2]…” This wording was confusing; did the authors mean “the combination of [Ab1] and [Ab2]”?

We replaced “association” by “combination” in the manuscript.

  • The sentence on lines 397-398 is repeated almost verbatim on lines 438-439.

The sentence on lines 438-439 was removed.

  • Please edit Figure 2 to more clearly indicate that in the “single batch manufacturing” scenario, a mixed culture of producer cell lines is used (perhaps make the oval shapes different colors). As is, it is almost identical with the schematic above.   

As requested, figure 2 was modified.

  • The black-on-blue format of the table headings makes them quite difficult to read, both in print and on screen

As requested, the color of table 1 was modified.
